# A novel incision design for Vibrant SoundBridge® implantation prior to auricular reconstruction for microtia

Naohiro Ueda[1]*, Takatoshi Yotsuyanagi[1], Ken Yamashita[1], Hidekane Yoshimura[2], Yutaka Takumi[2], Masahiro Takahashi[3], Satoshi Iwasaki[3], Shunsuke Yuzuriha[4], Shin-ichi Usami[5]

1 Department of Plastic and Reconstructive Surgery, Sapporo Medical University School of Medicine, Sapporo, Hokkaido, Japan, 2 Department of Otorhinolaryngology, Shinshu University School of Medicine, Matsumoto, Nagano, Japan, 3 Department of Otorhinolaryngology, International University of Health and Welfare Mita Hospital, Tokyo, Japan, 4 Department of Plastic and Reconstructive Surgery, Shinshu University School of Medicine, Matsumoto, Nagano, Japan, 5 Department of Hearing Implant Sciences, Shinshu University School of Medicine, Matsumoto, Nagano, Japan

* nueda@sapmed.ac.jp

**Data availability statement:** All relevant data are within the manuscript.

**Funding:** This work was supported by JSPS KAKENHI Grant Number JP23K09083.The

## Abstract

### Objective

This study aimed to present and evaluate a surgical approach for placing a Vibrant Soundbridge® (VSB) device prior to auricular reconstruction in patients with microtia atresia. The goal was to determine whether an incision line for VSB implantation, located sufficiently posterior to the planned area of auricular reconstruction, could prevent interference with subsequent auricular surgeries and ensure stable hearing outcomes.

### Methods

We retrospectively examined four patients with unilateral microtia who underwent VSB implantation before auricular reconstruction. The incision line was placed approximately 5 cm posterior to the posterior edge of the temporomandibular joint, ensuring that it remained at least 2 cm away from the future area of auricular reconstruction. The VSB implantation procedure involved skin incision, subperiosteal dissection, mastoidectomy, and placement of the VSB implant, with a floating mass transducer attached to the head of the stapes. More than a year later, two-stage auricular reconstruction was performed.

### Results

All patients underwent successful auricular reconstruction after VSB implantation. During the first stage of costal cartilage grafting, subcutaneous pockets were created without encountering congestion or compromised skin flaps, and the grafted cartilage maintained a good contour. In the second stage of auricular elevation, the mastoid

funder had no role in study design, data collection and analysis, decision to publish, or preparation of the manuscript.

**Competing interests:** The authors have declared that no competing interests exist.

fascial flap was elevated with adequate blood flow, and careful dissection prevented exposure or damage to the VSB device. Postoperatively, the reconstructed auricles exhibited stable contour. Audiometric evaluation revealed no deterioration in hearing outcomes, indicating that the VSB device worked well.

## Conclusion

This method allows safe and effective VSB implantation prior to auricular reconstruction, without hindering subsequent procedures. By maintaining an appropriate distance between the incision line and planned area of reconstruction, early hearing rehabilitation and successful auricular reconstruction can be achieved. This approach may serve as a practical standard for treating patients with microtia atresia who require early hearing intervention.

## Introduction

Most patients with microtia have congenital external auditory canal atresia and conductive hearing loss. These patients require hearing devices, including non-surgical options such as bone conduction hearing aids, adhesive hearing aids (Adhear), and cartilage conduction hearing aids, as well as surgical options such as bone conduction implants (Baha®, Osia®, and Bonebridge®) and middle ear implants (Vibrant Soundbridge® [VSB]). Among these options, VSB can improve hearing in the impaired ear without affecting the contralateral ear in cases of unilateral microtia atresia, suggesting that VSB may be an ideal treatment in such cases [1].

Since the 1980s, delays in language development and learning have been reported even in patients with unilateral microtia atresia. However, active hearing interventions for unilateral microtia atresia have not been widely implemented [2,3]. Recently, improvements in hearing using VSB have been reported not only bilaterally but also unilaterally, suggesting its potential for more proactive applications in the future [4,5]. Early initiation of hearing improvement is desirable, and it is recommended to perform VSB implantation before auricular reconstruction [1,6]. In such cases, careful consideration is required to ensure that the incision line used in VSB implantation does not adversely affect future auricular reconstructions.

We collaborated preoperatively with plastic surgeons and otolaryngologists to establish an incision line that allowed both surgeries to proceed smoothly. Using this incision line, we successfully performed a series of procedures involving VSB implantation and auricular reconstruction, without any critical complications. Here, we report the first detailed case of auricular reconstruction performed after VSB implantation.

## Materials and methods

This method can be used for patients with bilateral or unilateral microtia. The incision line was designed as a curved incision located 5 cm posterior to the posterior edge of the temporomandibular joint, with cranial extension slightly away from the temporomandibular joint (Fig 1). With this design, the incision line is generally

positioned > 2 cm away from the planned area of auricular reconstruction. The surgical technique for VSB implantation using this incision line has been previously reported by Yoshimura et al. [6].

The skin was incised directly down to the periosteum and the skin flap was elevated by dissecting above the bone. The temporal bone was exposed, and mastoidectomy was performed. Finally, the implant was placed on the temporal bone and a floating mass transducer was attached to the head of the stapes. More than a year after VSB implantation, two-stage auricular reconstruction was performed [7–9]. In the first stage, costal cartilage from the 6th to 8th ribs was harvested to create a cartilage framework, which was then subcutaneously grafted. After an interval of at least 6 months, the second-stage surgery involved elevating the auricle. The base of the auricle created in the first stage was in contact with the temporoparietal fascial flap (TPF), and the auricle was elevated, including the TPF. A costal cartilage strut was formed and grafted onto the posterior side of the auricle, and a superficial mastoid fascial flap was used for coverage. Finally, a full-thickness skin graft from the lower abdomen was placed onto the mastoid fascial flap to complete a series of surgeries. This study was approved by the Institutional Review Board of Sapporo Medical University Hospital. Written informed consent for all surgical procedures was obtained from the patients' guardians. All individuals pictured in Figs 1–3 and 5 have provided written informed consent (as outlined in PLOS consent form) to publish their image alongside the manuscript.

This retrospective study involved accessing medical records to collect data from December 9 to December 22, 2024. During data collection, personally identifiable information—such as patient names, dates of birth, and photographs—was accessible. However, following collection, the data were anonymized by assigning patient numbers to prevent individual identification.

## Results

Between 2021 and 2023, two-stage auricular reconstruction surgery following VSB implantation was performed in four patients at our institution (Table 1). All patients had unilateral microtia atresia; however, one patient had microtia associated with Treacher Collins syndrome and underwent bilateral VSB implantation. Representative cases are presented below.

### Patient 1

An 11-year-old boy with right lobule-type microtia underwent VSB implantation at the otorhinolaryngology department of Mita Hospital 1 year prior to costal cartilage grafting (Fig 2). During cartilage grafting, scars and implants from VSB implantation were observed posterior to the planned area of auricular reconstruction. A W-shaped skin incision was made

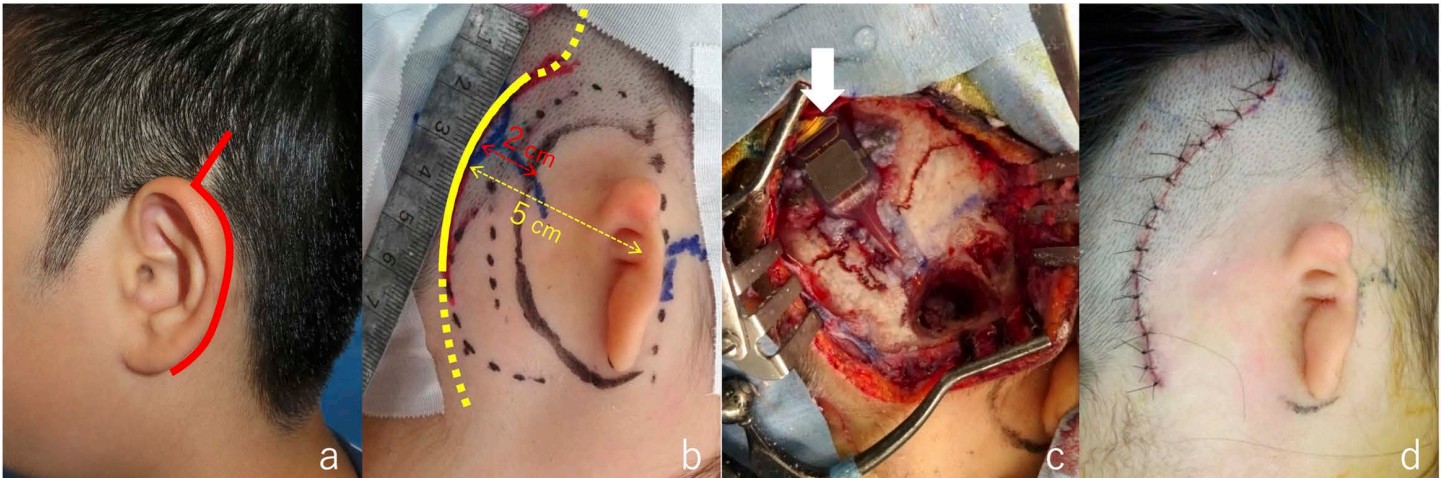

**Fig 1. Comparison between the conventional and novel incision lines.** (a) Conventional incision line for VSB surgery. (b) Our proposed incision line (solid line). The dotted line indicates the potential direction of extension. (c) VSB implantation (the arrow indicates the implant). (d) Post-suturing view.

**Table 1. Patient characteristics.**

| Patient | Sex | Age at VSB surgery | VSB Surgery Facility | Age at Costal Cartilage Graft | Type of Remnant Ear | Other Congenital Anomalies |
|---|---|---|---|---|---|---|
| 1 | M | 10 | Mita Hospital | 11 | Lobule | None |
| 2 | F | 8 | Mita Hospital | 11 | Lobule | None |
| 3 | M | 9 | Shinshu University Hospital | 11 | Lobule | None |
| 4 | M | 8* | Mita Hospital | 10 | Lobule | Treacher Collins Syndrome |

*Note: The recorded age at VSB implantation corresponds to the right side, for which auricular reconstruction was performed. On the left side, although there was no abnormality in auricular morphology, VSB implantation was performed at the age of 7 to improve hearing.

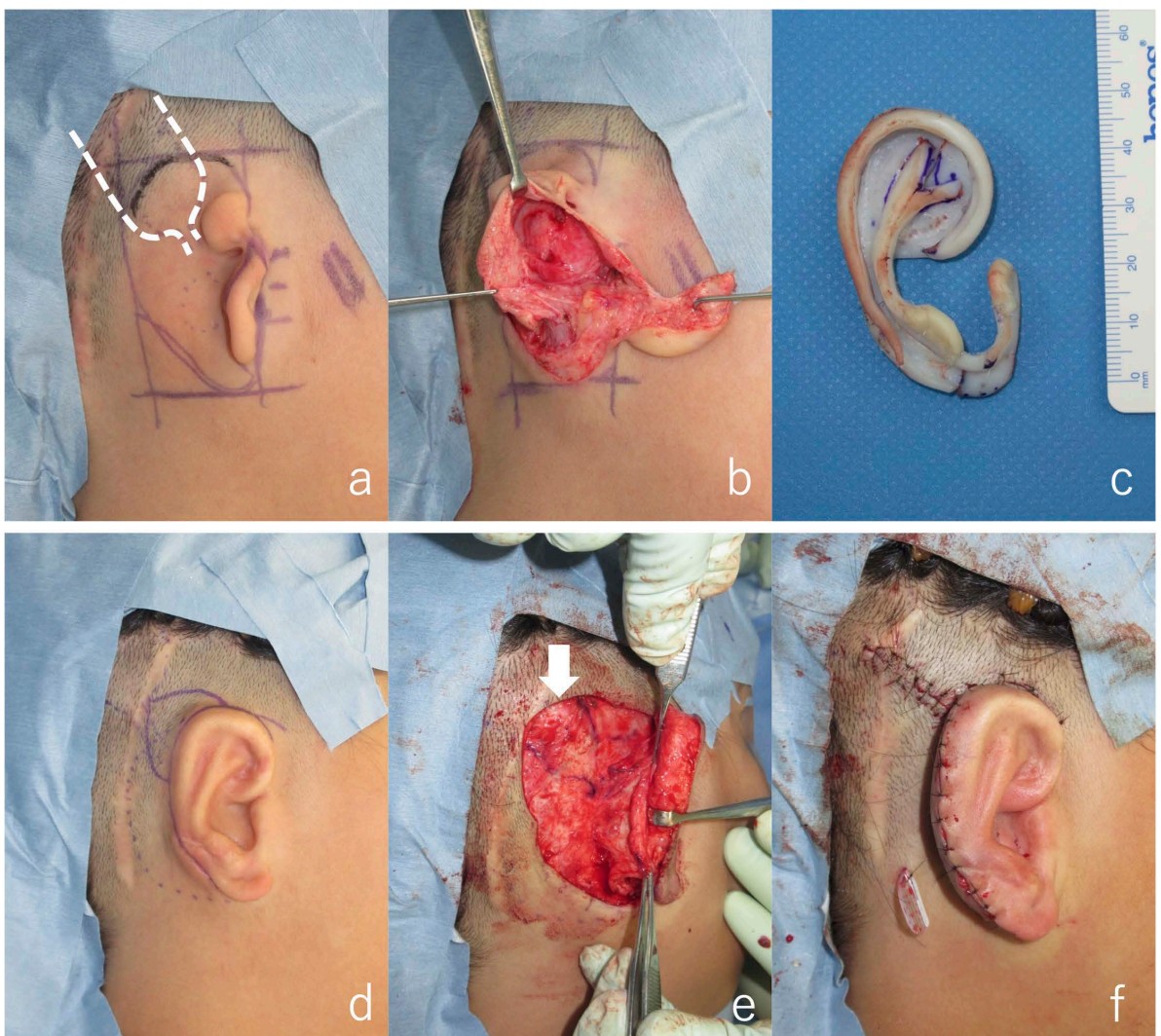

**Fig 2. Two-stage auricular reconstruction surgery after VSB implantation.** (a) Design of costal cartilage grafting: first stage (the dotted line indicates the implant location). (b) Creation of the subcutaneous pocket. (c) Costal cartilage framework for grafting. (d) Design of auricular elevation: second stage. (e) During mastoid fascial flap elevation (the arrow indicates the implant visible through the tissue). (f) After completion of auricular elevation.

[7,10]. A subcutaneous pocket was created through standard dissection, maintaining a distance of at least 1 cm between the pocket and VSB incision line. The surgery was performed using a typical costal cartilage graft. Postoperatively, the dissected skin healed well without congestion and the grafted cartilage maintained a good contour.

Eight months after cartilage grafting, auricular elevation was performed. An incision was made along the outer edge of the auricle for elevation, and a superficial mastoid fascial flap was raised to cover the cartilage strut. Although the implant was visible beneath the fascia, careful dissection prevented its exposure. The elevated fascia maintained good blood flow, and the grafted full-thickness skin survived. After 1 year and 4 months of follow-up, the auricular contour remained satisfactory (Fig 3). The surgical scar from VSB implantation was concealed by hair and was unnoticeable. Audiograms showed no change in hearing before or after auricular reconstruction, indicating that the device was not damaged during surgery (Fig 4).

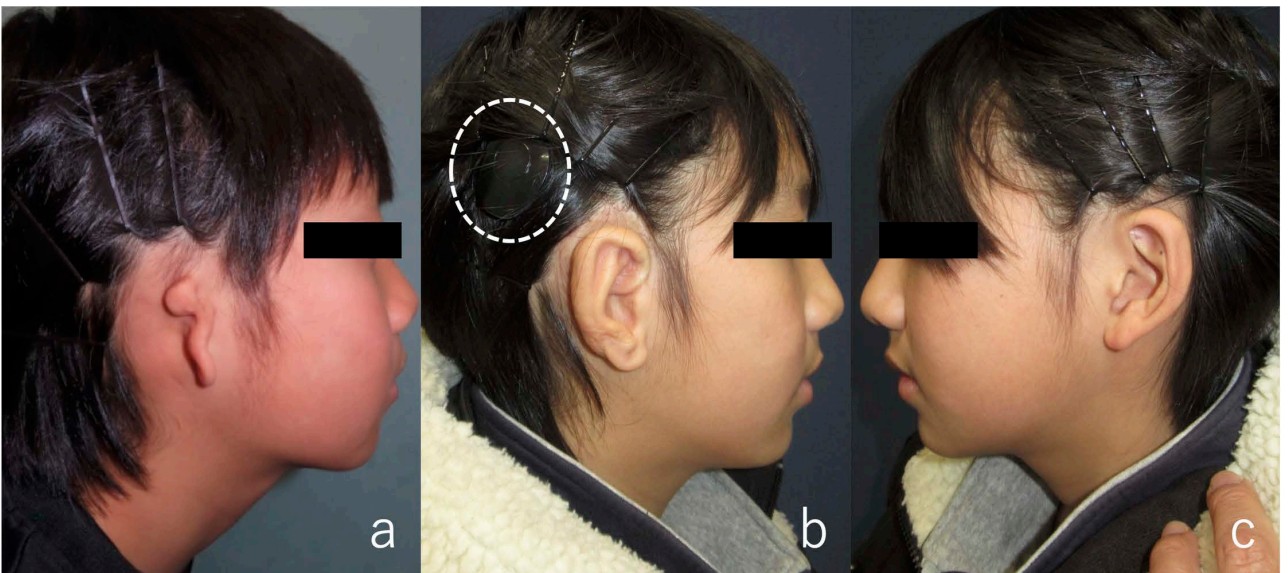

**Fig 3. Preoperative and postoperative contour of the auricle (Patient 1).** (a) Before auricular reconstruction on the right ear. (b), (c) One year and four months after auricular elevation surgery, maintaining a good contour. The dotted circles indicate the processor of VSB.

## Before auricular reconstruction
## After auricular reconstruction

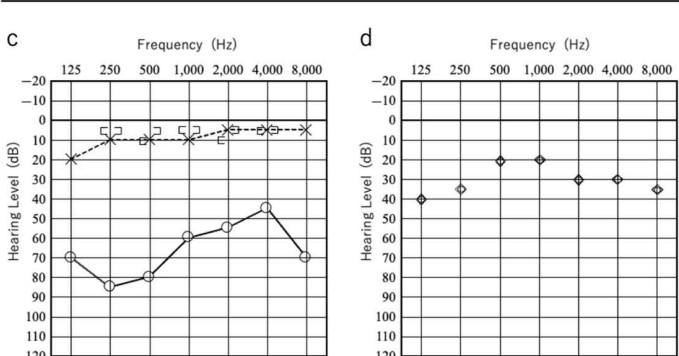

**Fig 4. Audiograms of before and after auricular reconstruction (Patient 1).** (a), (c) Pure-tone audiograms. (b), (d) Hearing thresholds with VSB in the right ear. No change in the hearing level was observed before or after the procedures. ○: Air conduction (right ear), ×: Bone conduction (left ear). [ : Bone conduction (right ear), ] : Bone conduction (left ear).

**Patient 2**

An 11-year-old girl with right lobule-type microtia underwent VSB implantation at the otorhinolaryngology department of Mita Hospital 3 years prior to costal cartilage grafting. Surgery was performed in a manner similar to that for Patient 1. Auricular elevation surgery was conducted 6 months after the initial cartilage grafting. In both surgeries, the postoperative course was uneventful and the contour was satisfactory (Fig 5). There were no significant changes in hearing before or after the surgery (Fig 6).

As shown in Fig 4, no changes in hearing were observed before or after auricular reconstruction.

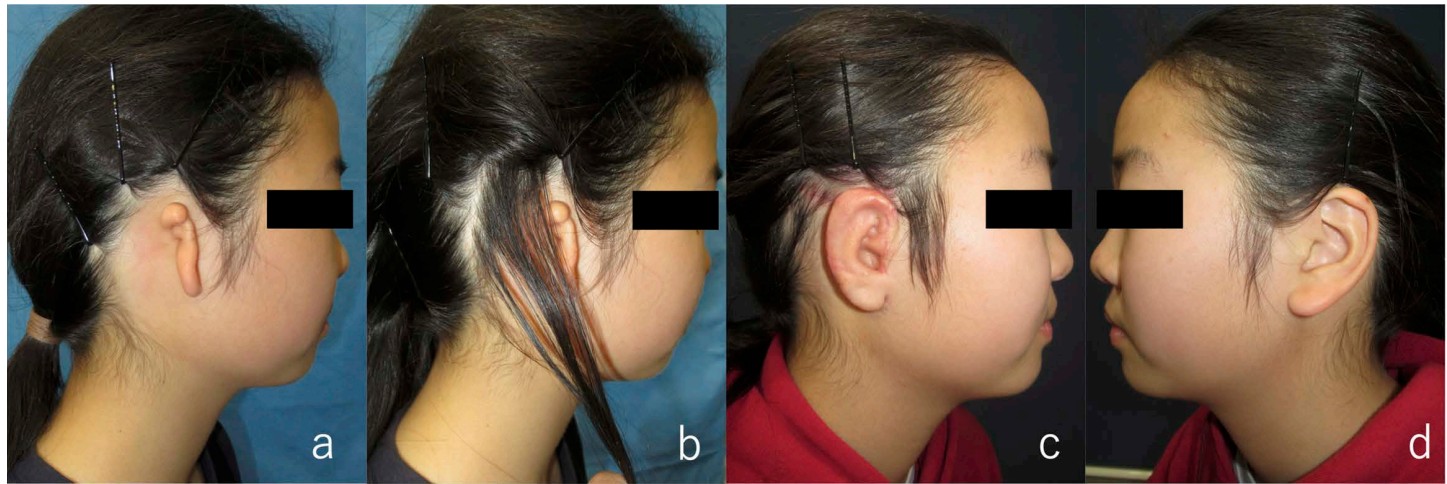

**Fig 5. Preoperative and postoperative contour of the auricle (Patient 2).** (a) Before auricular reconstruction on the right ear. (b) Post-VSB implantation scar. The scar can be concealed by hair. (c), (d) Three months after auricular elevation surgery. Although slight swelling persisted, the shape remained intact.

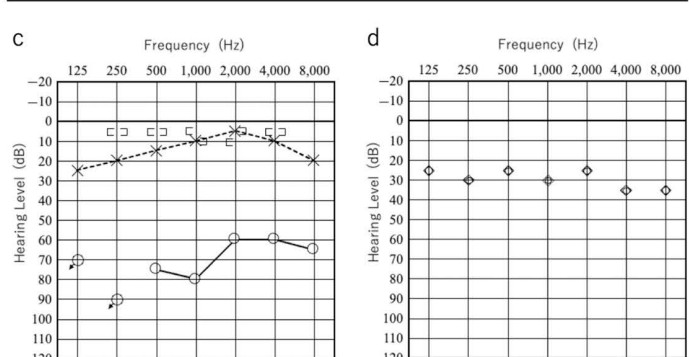

**Fig 6. Audiograms of before and after auricular reconstruction (Patient 2).** (a), (c) Pure-tone audiograms. (b), (d) Hearing thresholds with VSB in the right ear.

## Discussion

VSB has been increasingly used as an effective method for patients with normal inner ears but abnormalities in the outer or middle ear. Reports of VSB application in microtia patients have become more common, often involving VSB implantation performed after auricular reconstruction or simultaneously with auricular elevation [4,11–13]. However, reports on VSB implantation performed before auricular reconstruction are limited [14,15]. The use of VSB in patients with unilateral hearing loss improves speech understanding in noisy environments and sound localization abilities. In addition, early treatment is generally advocated for patients with congenital hearing loss or hearing loss in early childhood [1,6]. Therefore, it is important to consider performing VSB implantation before auricular reconstruction, which is typically done at 10–11 years of age [7].

The incision line commonly used in VSB implantation can significantly interfere with the subsequent auricular reconstruction. Therefore, designing an incision line that anticipates future auricular reconstruction is crucial. Auricular reconstruction is generally performed in two stages: the first involves costal cartilage grafting, and the second, conducted approximately 6 months later, involves auricular elevation. The incision line used during VSB implantation significantly affects both the stages. In the first cartilage graft, the subcutaneous dissection extends up to approximately 1 cm lateral to the planned area of cartilage grafting. Adequate blood flow from the surrounding skin is essential; thus, skin incisions that can compromise the blood flow should be avoided. Therefore, the incision line should be placed at least 2 cm away from the planned area of auricular reconstruction. During auricular elevation surgery, the Nagata method uses a TPF to cover the posterior surface of the auricle [10], whereas we used a mastoid fascial flap. Incisions that could impair the blood flow to these fascial flaps should be avoided. Preserving the TPF is also desirable for salvage procedures in cases of infection or cartilage exposure [16,17]. Considering these points, during VSB implantation, we made a skin incision 5 cm away from the external auditory canal. To minimize damage to the fascia, we incised it vertically from the incision line down to the periosteum and performed subperiosteal dissection. In our cases, no skin congestion occurred during the initial surgery, and stable blood flow was maintained in the fascia during auricular elevation. Although the implant was visible beneath the fascia during elevation, careful dissection prevented device exposure and damage.

Frenzel et al. reported a similar approach, in which VSB implantation preceded auricular reconstruction [14]. They determined the auricular position based on nasal and ocular landmarks by setting an incision line 15 mm away from the helix rim. Although they reported a successful auricular reconstruction, they did not provide postoperative images for evaluation. Their incision line extended anteriorly to the auricle, potentially damaging the superficial temporal artery that supplies blood to the TPF, which could complicate its use during auricular elevation. Our incision avoided this risk by extending cranially away from the auricle. Their method requires precise auricular positioning, ideally determined by the plastic surgeon performing the reconstruction. However, in Japan, only a limited number of surgeons specialize in microtia reconstruction, and they are often not affiliated with the same institutions as otolaryngologists. Our method allows otolaryngologists to easily set the incision line without the need for a plastic surgeon.

Our study had several limitations. First, because the incision line was significantly more posterior than the position of the external auditory canal, there was a concern that the dissection required for VSB implantation might become more complicated. However, Yoshimura et al. reported successful VSB implantation using this incision without complications [6]. Second, there were concerns regarding the aesthetic impact of a long scar without hair growth behind the auricle. In practice, the scar was concealed by hair and the reconstructed auricle, posing no significant aesthetic problems. Third, although no flap necrosis or device damage was observed in this study, the small sample size precluded statistical analysis, and further case accumulation is warranted for future evaluation.

We believe that our method is a suitable standard approach for treating patients with relatively symmetrical facial features. However, for patients with hemifacial microsomia, in whom temporomandibular joint anomalies may be present, careful consideration of incision placement is necessary. Various reconstructive techniques have been described for

patients with acquired auricular defects following tumor resection or trauma [18–20]. We believe our method is unsuitable for these patients. In these cases, VSB implantation is generally not performed prior to auricular reconstruction. Instead, it is technically simpler and preferable to implant the VSB either after auricular reconstruction or concurrently with auricular elevation [4,11–13].

## Conclusion

To the best of our knowledge, this is the first detailed report of auricular reconstruction following VSB implantation. Our method successfully improved both hearing and auricular contour. Although ideally performed by a plastic surgeon, our incision line allows otolaryngologists to perform the surgery independently. We anticipate broader adoption of VSB implantation in the future.

## Acknowledgments

We would like to thank Editage (www.editage.jp) for English language editing.

## Author contributions

**Conceptualization:** Naohiro Ueda, Takatoshi Yotsuyanagi, Hidekane Yoshimura, Yutaka Takumi, Masahiro Takahashi, Satoshi Iwasaki, Shunsuke Yuzuriha, Shin-ichi Usami.

**Data curation:** Naohiro Ueda.

**Funding acquisition:** Naohiro Ueda.

**Investigation:** Naohiro Ueda, Takatoshi Yotsuyanagi, Ken Yamashita.

**Methodology:** Takatoshi Yotsuyanagi, Hidekane Yoshimura, Yutaka Takumi, Masahiro Takahashi, Satoshi Iwasaki, Shunsuke Yuzuriha, Shin-ichi Usami.

**Supervision:** Takatoshi Yotsuyanagi, Satoshi Iwasaki, Shunsuke Yuzuriha, Shin-ichi Usami.

**Visualization:** Naohiro Ueda, Hidekane Yoshimura, Masahiro Takahashi.

**Writing – original draft:** Naohiro Ueda, Hidekane Yoshimura.

**Writing – review & editing:** Takatoshi Yotsuyanagi, Ken Yamashita, Hidekane Yoshimura, Yutaka Takumi, Masahiro Takahashi, Satoshi Iwasaki, Shunsuke Yuzuriha, Shin-ichi Usami.

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
