## [Decision Letter · Decision Letter 0]

PONE-D-25-13505
A novel incision design for Vibrant SoundBridge®︎ implantation prior to auricular reconstruction for microtia
PLOS ONE

Dear Dr. Ueda,

Thank you for submitting your manuscript to PLOS ONE. After careful consideration, we feel that it has merit but does not fully meet PLOS ONE’s publication criteria as it currently stands. Therefore, we invite you to submit a revised version of the manuscript that addresses the points raised during the review process.

We look forward to receiving your revised manuscript.

Kind regards,

Shimpei Miyamoto

Academic Editor

PLOS ONE

Journal Requirements:

Reviewers' comments:

Reviewer's Responses to Questions

**Comments to the Author**

1. Is the manuscript technically sound, and do the data support the conclusions?

Reviewer #1: Yes

Reviewer #2: Partly

2. Has the statistical analysis been performed appropriately and rigorously? 

Reviewer #1: N/A

Reviewer #2: Yes

3. Have the authors made all data underlying the findings in their manuscript fully available?

Reviewer #1: Yes

Reviewer #2: Yes

4. Is the manuscript presented in an intelligible fashion and written in standard English?

Reviewer #1: Yes

Reviewer #2: Yes

5. Review Comments to the Author

Reviewer #1: At line 139

"A subcutaneous pocket was created through standard dissection"

You did not mention which incision was used to access the pocket.

In table 1 last column

It us other congenital anomalies and not other complications

In figure 5a the scar of insertion of VSB is not visible

Please add to the limitation of the study small number of cases

Reviewer #2: Many thanks for submitting this work to APS.

Authors should just enrich the "discussion" section by briefly mention two concepts:

1) the role of technology for the assessment of prognosis in contemporary Plastic Surgery (for every field of this Discipline, from wounds to microsurgery);

2) the potential future expansion of this kind of method for post oncologic

auricular reconstruction.

Please, cite the following papers

- Guarro G, Cozzani F, Rossini M, Bonati E, Del Rio P. Wounds morphologic assessment: application and reproducibility of a virtual measuring system, pilot study. Acta Biomed. 2021 Nov 3;92(5):e2021227. doi: 10.23750/abm.v92i5.11179. PMID: 34738578; PMCID: PMC8689305.

-Boissiere F, Gandolfi S, Riot S, Kerfant N, Jenzeri A, Hendriks S, Grolleau JL, Khechimi M, Herlin C, Chaput B. Flap Venous Congestion and Salvage Techniques: A Systematic Literature Review. Plast Reconstr Surg Glob Open. 2021 Jan 22;9(1):e3327. doi: 10.1097/GOX.0000000000003327. PMID: 33564571; PMCID: PMC7858245

- Guarro G, Fabrizio T. Indications for and Limitations of Reconstruction of Auricular Defects with the "Mid-moon Flap" and Evaluation of Outcome by the Aesthetic Numeric Analogue Score. Plast Reconstr Surg Glob Open. 2023 Jul 25;11(7):e5152. doi: 10.1097/GOX.0000000000005152. PMID: 37496980; PMCID: PMC10368381.

- Moreno-Vazquez S, Antoñanzas J, Oteiza-Rius I, Redondo P, Salido-Vallejo R. Reconstructive Procedures of the Auricular Concha after Cutaneous Oncologic Surgery: A Systematic Review. J Clin Med. 2023 Oct 14;12(20):6521. doi: 10.3390/jcm12206521. PMID: 37892659; PMCID: PMC10607053.

- Guarro G, Cozzani F, Rossini M, Bonati E, Del Rio P. The modified TIME-H scoring system, a versatile tool in wound management practice: a preliminary report. Acta Biomed. 2021 Sep 2;92(4):e2021226. doi: 10.23750/abm.v92i4.10666. PMID: 34487096; PMCID: PMC8477093.

Many thanks.

6. PLOS authors have the option to publish the peer review history of their article (what does this mean?). If published, this will include your full peer review and any attached files.

Reviewer #1: **Yes: **Ahmed Elshahat

Reviewer #2: No

---

## [Author Response · Author response to Decision Letter 1]

25 May 2025

Reviewer #1:

Thank you very much for your thoughtful comments and questions. We have addressed each of them below.

At line 139

"A subcutaneous pocket was created through standard dissection"

You did not mention which incision was used to access the pocket.

→We have added the following text to the main body of the manuscript.

“A W-shaped skin incision was made.”

In table 1 last column

It is other congenital anomalies and not other complications

→We have revised the wording from “other complications” to “other congenital anomalies.”

In figure 5a the scar of insertion of VSB is not visible

→We have added a photograph that clearly shows the scar from the VSB implantation, now presented as Fig. 5b.

Please add to the limitation of the study small number of cases

→We have added the following text to the main body of the manuscript.

“Third, although no flap necrosis or device damage was observed in this study, the small sample size precluded statistical analysis, and further case accumulation is warranted for future evaluation.”

Reviewer #2:

Authors should just enrich the "discussion" section by briefly mention two concepts:

1) the role of technology for the assessment of prognosis in contemporary Plastic Surgery (for every field of this Discipline, from wounds to microsurgery)

→ Thank you for suggesting those references. We have carefully reviewed the papers you recommended, which describe methods for ulcer assessment using devices and scoring systems. These are indeed highly interesting contributions to the field of wound management. While enabling prognostic evaluation of wounds is undoubtedly an important challenge in wound care, it lies beyond the primary scope of our current manuscript. Therefore, we have decided not to include additional discussion on this topic.

2) the potential future expansion of this kind of method for post oncologic

auricular reconstruction.

→ Thank you for your valuable comments. We have added the following text to the manuscript to address the applicability of our method to acquired auricular defects, including those resulting from tumor resection or trauma. We have also cited the references you kindly suggested.

“Various reconstructive techniques have been described for patients with acquired auricular defects following tumor resection or trauma. We believe our method is unsuitable for these patients. In these cases, VSB implantation is generally not performed prior to auricular reconstruction. Instead, it is technically simpler and preferable to implant the VSB either after auricular reconstruction or concurrently with auricular elevation.”

---

## [Editor Report · Decision Letter 1]

A novel incision design for Vibrant SoundBridge®︎ implantation prior to auricular reconstruction for microtia

PONE-D-25-13505R1

Dear Dr. Ueda,

We’re pleased to inform you that your manuscript has been judged scientifically suitable for publication and will be formally accepted for publication once it meets all outstanding technical requirements.

Kind regards,

Shimpei Miyamoto

Academic Editor

PLOS ONE
---

## [Editor Report · Acceptance letter]

PONE-D-25-13505R1

PLOS ONE

Dear Dr. Ueda,

I'm pleased to inform you that your manuscript has been deemed suitable for publication in PLOS ONE. Congratulations! Your manuscript is now being handed over to our production team.

Kind regards,

on behalf of

Dr. Shimpei Miyamoto

Academic Editor

PLOS ONE